# First measurements of p¹¹B fusion in a magnetically confined plasma

R. M. Magee [1] ✉, K. Ogawa [2], T. Tajima[1,3], I. Allfrey [1], H. Gota [1], P. McCarroll[1], S. Ohdachi [2], M. Isobe[2], S. Kamio [1,3], V. Klumper[1,3], H. Nuga[2], M. Shoji[2], S. Ziaei[1], M. W. Binderbauer[1] & M. Osakabe [2]

Proton-boron (p¹¹B) fusion is an attractive potential energy source but technically challenging to implement. Developing techniques to realize its potential requires first developing the experimental capability to produce p¹¹B fusion in the magnetically-confined, thermonuclear plasma environment. Here we report clear experimental measurements supported by simulation of p¹¹B fusion with high-energy neutral beams and boron powder injection in a high-temperature fusion plasma (the Large Helical Device) that have resulted in diagnostically significant levels of alpha particle emission. The injection of boron powder into the plasma edge results in boron accumulation in the core. Three 2 MW, 160 kV hydrogen neutral beam injectors create a large population of well-confined, high-energy protons to react with the boron plasma. The fusion products, MeV alpha particles, are measured with a custom designed particle detector which gives a fusion rate in very good relative agreement with calculations of the global rate. This is the first such realization of p¹¹B fusion in a magnetically confined plasma.

The proton–boron fusion reaction (p¹¹B),

$$\mathrm{p}(^{11}\mathrm{B}, \alpha)\alpha\alpha + 8.7\,\mathrm{MeV} \tag{1}$$

has long been recognized as attractive for fusion energy[1]. The reactants, hydrogen and boron, are abundant in nature, non-toxic and non-radioactive, and the reaction itself produces no neutrons, only helium in the form of three alpha particles. There are challenges, most notably that the temperature required for a thermonuclear p¹¹B fusion reactor is 30 times higher than that for deuterium-tritium (DT), the candidate fusion fuel with the lowest operating temperature. Since in a plasma, high temperature typically means large radiated power in the form of synchrotron and bremsstrahlung radiation, this makes finding an operating point in which the fusion output power is greater than the input power more challenging. Because of this, research groups pursuing p¹¹B for magnetic fusion energy remain a small minority worldwide.

While the challenges of producing the fusion core are greater for p¹¹B than DT, the engineering of the reactor will be far simpler. The enormous fluence of 14 MeV neutrons from a DT reactor plasma (~$10^{19}$ n/m²/s) will require advanced, yet-to-be-developed materials for the first wall, threaten the integrity of superconducting coils, and necessitate remote handling of activated materials. None of these concerns apply to a reactor based on the aneutronic p¹¹B reaction. Stated simply, the p¹¹B path to fusion trades downstream engineering challenges for present day physics challenges.

And the physics challenges can be overcome. As demonstrated in ref. [2], by using the recently updated values for the p¹¹B fusion cross-section[3] and properly accounting for kinetic effects, it can be shown that a thermal p¹¹B plasma can produce a high $Q$ (where $Q$ = fusion power/input power), and even reach ignition (where the plasma is sustained by the fusion reactions alone). By employing a plasma with a low internal magnetic field and operating in a regime in which the electrons are kept at a lower temperature than the ions, the radiation

¹TAE Technologies, Inc., Foothill Ranch, CA, USA. ²National Institute for Fusion Science, Toki, Japan. ³University of California—Irvine, Irvine, CA, USA. ✉ e-mail: RMagee@TAE.com

losses can be further reduced[1]; and by maintaining a non-equilibrium population of energetic reacting ions, the fusion power further increased[4]. To this end, TAE is developing the ideas first put forward by Rostoker[5] with the beam-driven, field reversed configuration (FRC). In this concept, the naturally high-beta FRC plasma[6,7] serves as both container and fusion target for a large, neutral beam-injected fast ion population.

Proton–boron fusion has been studied theoretically[8,9], in laser produced plasmas[10], and in particle accelerators through "beam-target fusion"[3,11], but there has not been, until now, the opportunity to study it in a magnetically confined fusion plasma. In beam-target fusion, a particle beam is impingent on a solid target and the interactions between the beam and target particles are limited to binary collisions and short-range interactions. The magnetically confined plasma is a much richer environment, allowing for studies of collective effects on the fusion reaction rate. For example, recent work on wave-particle interaction in the FRC indicate a natural boost to the fusion output due to a beam-driven wave[12,13]. The effects of the resonances in the p[11]B cross section in the presence of high-energy tails and non-thermal distributions on the fusion rate can also be explored. In short, the present work marks the beginning of experimental studies of p[11]B fusion in beam-driven systems, an important milestone in the development of fusion energy.

The work described below is the result of a private-public partnership between the National Institute for Fusion Science in Japan (NIFS) and TAE and builds on a long history of US-Japan collaboration in fusion energy research[14]. In the following Article, we will describe how the experimental capabilities of the LHD were utilized to produce p[11]B alphas and how we diagnose them. We then compare the measured count rate to a global p[11]B fusion reaction rate calculation, and, finally, describe future work.

## Results
### Producing fusion alpha particles in LHD
In order to produce the target boron plasma, we utilize the boronization system of the LHD. Boronization is a standard wall conditioning technique in magnetic fusion devices. It can be done via glow discharge between plasma shots, or, as is the case of the LHD system, in real time. Real time boronization improves plasma performance through both indirect and direct actions. Boron is delivered to the plasma as sub-millimeter grains of pure boron or boron nitride (BN) with the Impurity Powder Dropper (IPD)[15,16], a system designed, built and installed on the LHD by Princeton Plasma Physics Laboratory. It conditions the walls of the vessel to decrease recycling and intrinsic impurity content (e.g., C, O, Fe). This leads to a decrease in turbulence and an improvement in global confinement[17,18]. It also acts as a supplemental electron source, increasing electron density and thus neutral beam (NB) deposition. Most relevant to the current study, charge exchange recombination spectroscopy measurements reveal that a significant amount of boron accumulates in the mid-radius of the

plasma during boron powder injection, leading to boron densities of up to $6 \times 10^{17} \text{m}^{-3}$.

The second critical ingredient, high-energy protons, are delivered with NB Injection. The LHD is equipped with a suite of NBs: two radially injected, positive ion source beams with energy $E = 60-80$ kV, and three negative ion source, tangential beams with $E = 135-180$ kV[19]. The few Tesla heliotron magnetic field provides good confinement of the beam-injected fast ions[20]. The tangential beams can therefore access the energies required for p[11]B fusion near the first resonance[21].

Calculations predict that the experimentally achieved boron densities and NB parameters will result in p[11]B fusion rates of ~$10^{14}\text{s}^{-1}$ when all three high-energy beams are fired simultaneously[22]. We also note that the large Larmor radius (~10 cm) of the high-energy alphas results in most being lost to the vessel wall or divertor plates in a few orbits, so that a diagnostic situated outside of the plasma can register a signal.

### Detecting alpha particles
The principal component of the alpha particle detector is a Passivated Implanted Planar Silicon (PIPS) detector from Mirion Technologies (PD 2000-40-300 AM). The PIPS detector is a large area (2000 mm²), partially depleted, silicon semiconductor operated in photodiode mode with 72 V reverse bias. Semiconductor detectors have been used on other magnetic fusion devices to detect fusion products[23–25].

The signal pulse arises when a photon or charged particle enters the depletion region of the semiconductor and creates a population of electron-hole pairs. These charge carriers give rise to a current that is collected and amplified with electronics external to the vacuum vessel. The detector is oriented such that there is no direct line-of-sight from the core plasma to the detector to minimze X-ray contamination; we rely on the ~3 T magnetic field of the LHD to steer the alpha particles to the detector, as depicted in Fig. 1. Lorentz orbit tracing calculations[22] were carried out to determine the orientation of the detector that maximizes alpha particle collection without directly facing the plasma. We also employ a thin foil (2 μm platinum) to shield stray X-rays, see the Methods section for details.

### Measurements
Plotted in Fig. 2 are data from a hydrogen discharge with toroidal magnetic field $B_t = 2.75$ T, major radius $R_{ax} = 3.60$ m, central electron temperature $T_e$ ~2 keV, and line averaged electron density $\langle n_e \rangle$ ~$2 \times 10^{18}\text{m}^{-3}$. The top plot in the left column shows that one high-energy hydrogen NB turns on at $t = 5.31$ s, while the bottom plot shows the signal on the alpha particle detector. It can be seen that at the time of beam turn-on, fast, negative spikes, with amplitude ~150 mV and a pulse shape consistent with that from calibration data taken with a [241]Am source appear on the detector (see Methods for details on the calibration).

The right column of Fig. 2 compares two otherwise identical discharges, one with boron injection and one without. In both discharges,

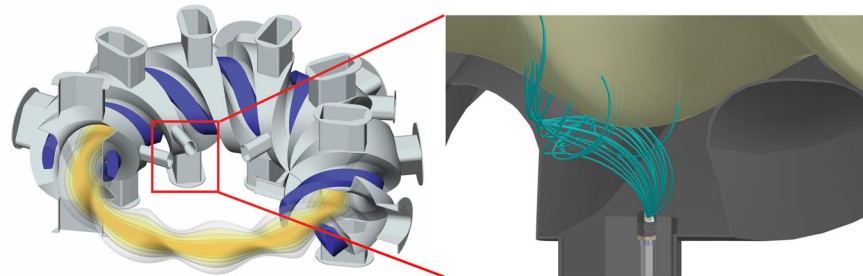

**Fig. 1 | Experimental set-up.** (Left) 3D CAD model showing the LHD vacuum vessel with cut-away view of heliotron plasma. (Right) CAD image showing calculated alpha particle trajectories (green curves) reaching the PIPS detector near the LHD separatrix, a portion of the last closed flux surface (tan), and the PIPS detector, located below the plasma.

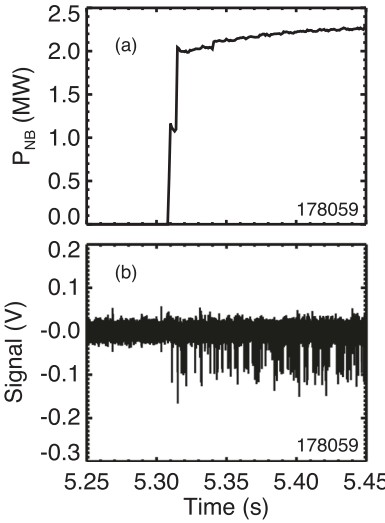

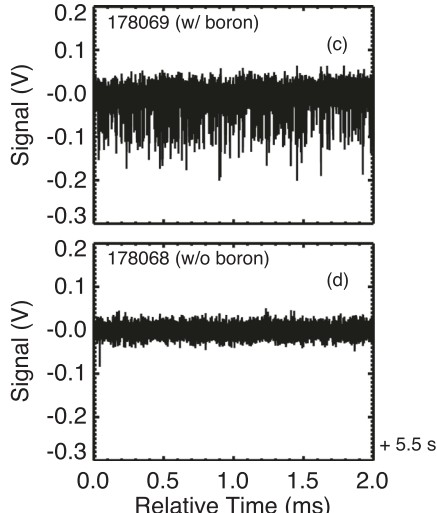

**Fig. 2 | Correlation of p$^{11}$B fusion alpha particle signal with NB and boron powder injection.** Left column shows the time history of the NB power (**a**) and the signal from the alpha particle detector (**b**). It can be seen that the negative pulses begin coincident with NB turn-on. Right column compares shots with (**c**) and without (**d**) boron powder injection. The pulse rate is dramatically reduced when boron is not injected. Note time scale differences, left column in seconds, right column in milliseconds. In these plasmas, the toroidal magnetic field $B_t = 2.75$ T, $R_{ax} = 3.60$ m, the central electron temperature $T_e$ -2 keV, and the line averaged electron density $\langle n_e \rangle$ - $2 \times 10^{18}$m$^{-3}$.

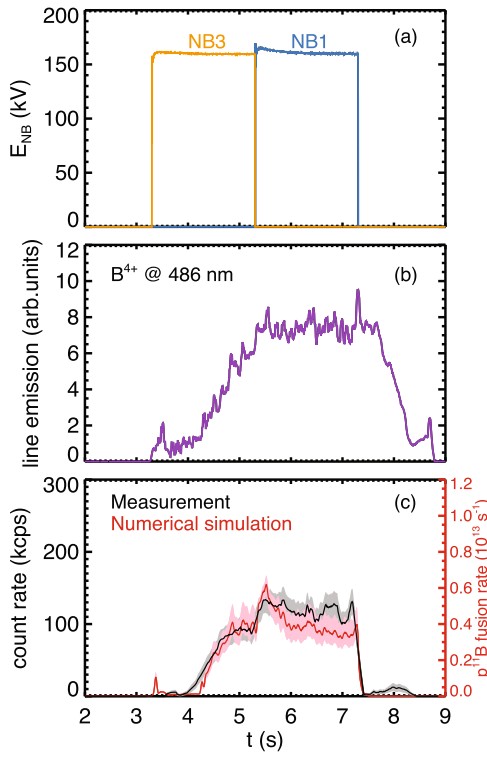

**Fig. 3 | Comparison of the measured count rate and global emission rate from numerical simulation. a** The voltage of the accelerating grid on tangential neutral beams 1 and 3 as a function of time. **b** Time evolution of line-integrated $B^{4+}$ intensity reflecting the amount of boron in the plasma. **c** Alpha particle detector pulse count rate (black) with error bars due to Poisson counting errors (gray) and calculated p$^{11}$B fusion rate (red) with error bars due to propagated uncertainty in beam energy (pink), scaled to match near $t = 5$ s.

NB3 was fired from 3.3 to 5.3 s and NB1 was fired from 5.3 to 7.3 s (as in Fig. 3a), so there is steady hydrogen NB injection during the window shown. It can be seen that the pulse rate is dramatically reduced in the case without boron injection. The maximum pulse count rate in the boron injected case is about 150 kcps, while in the case without it is less than 1 kcps. The finite count rate in the no boron case is likely due to residual boron in the plasma that had been deposited on the walls during previous shots.

Thus, the appearance of the signal pulses is clearly correlated with both the presence of boron and high-energy NB injection.

## Comparison to numerical simulation

Next, we compare the dynamics of the observed pulse count rate to simulation. The FBURN code[26] calculates the global p$^{11}$B fusion rate using experimentally measured inputs: NB injection power, bulk plasma parameters, boron density profile inferred from charge exchange recombination spectroscopy, and relative boron density from Extreme Ultra Violet (EUV) spectrometer. While the calculation of the fusion rate is straightforward, capturing the fast ion dynamics and the resulting fast ion profile and energy spectrum requires modeling[22].

In Fig. 3, the acceleration voltage of two co-injected, tangential NBs is plotted as a function of time showing that each was operated at 160 kV beginning at $t = 3.3$ s for a total of 4 s. In the middle frame is the line-integrated $B^{4+}$ intensity. Boron is injected by the IPD beginning at $t = 4.0$ s. In the bottom frame, the observed pulse count rate (black) and simulated global p$^{11}$B fusion rate (red) are plotted.

It can be seen that the slope of the leading edge in the measured pulse count rate tracks the rise in calculated rate well. This rate is governed by the boron accumulation time. At the end of the NB discharge at $t = 7.3$ s, the pulse count rate drops very quickly even though there is still boron in the plasma. It is therefore the beam ions that govern the dynamics of the curves' trailing edges. The error bars on the numerical simulation curve correspond to the effect of the $\pm 1$ kV uncertainty in the measured beam energy, and the error bars in the measurement are the Poisson counting errors in the chosen time window of 10 ms.

An absolute comparison between the measurement and calculation requires knowledge of the alpha energy spectrum at the source (i.e., in the plasma) for accurate orbit tracing, while we can only measure the energy spectrum of those alphas that arrive at the detector, creating severe survivorship bias. We therefore rely on the dynamical, relative agreement as further confirmation that the measured signal is indeed p$^{11}$B fusion born alpha particles.

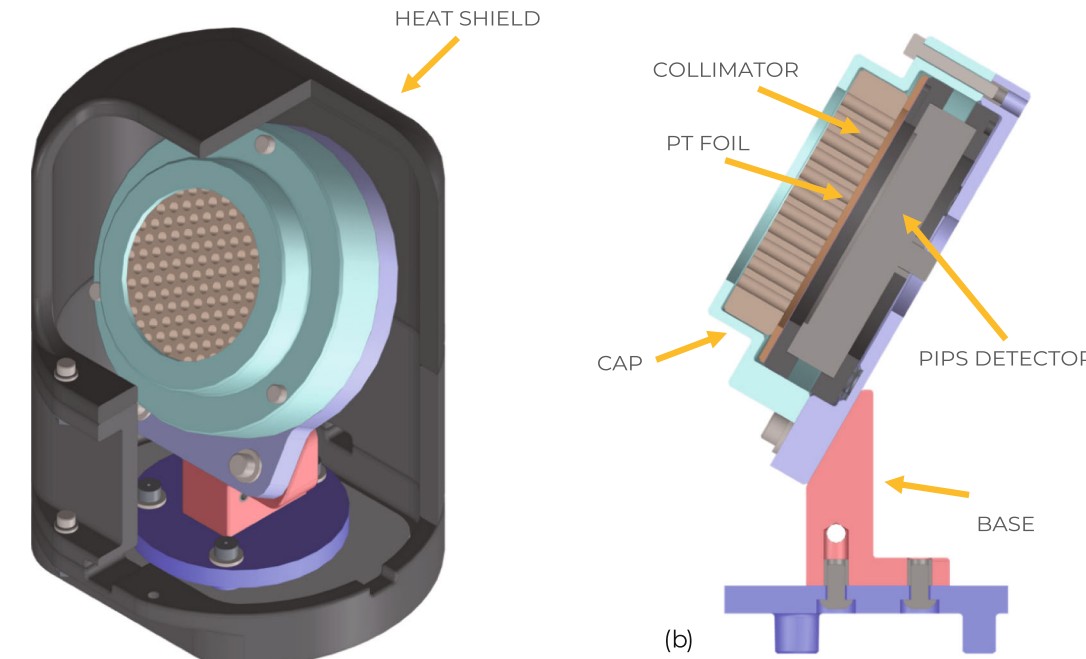

**Fig. 4 | Alpha particle detector. a** CAD model of the alpha particle detector with graphite heat shield partially cut-away. **b** Bisected view with principal components labeled. Collimator and Pt foil shield the PIPS detector from photons and low energy and particles.

## Discussion

The potential advantages of p[11]B over other fusion fuels are undeniable. The fuel is abundant, non-toxic, and non-radioactive, and there are no neutrons in the primary reaction, mitigating activation concerns. And, as mentioned above, it has been shown that with proper parameter tuning the energetics are favorable[2].

For the last several decades, the study of the p[11]B fusion reaction has been confined to theoretical reactor studies, nuclear physics experiments at particle accelerators, and laser driven plasma experiments. In order to develop practical techniques to truly enable magnetic fusion with p[11]B fuel, it will be critical to study the reaction in the environment in which it will be employed, a magnetically confined, thermonuclear plasma.

This will include developing techniques to further increase the fusion gain via alpha channeling[27], profile tuning, phase space engineering, fusion product current drive[14], and the exploitation of collective beam-induced heating[12,13]. It will also, as described above, involve studying the interplay of fast ion diffusion on the fusion rate in the presence of resonant peaks, something that can't be reproduced with fusion in a DD or DT plasma or in p[11]B beam-target fusion. The present work opens the door to these studies.

## Methods

### Diagnostic

The alpha particle detector is comprised of a 2000 mm², partially depleted semiconductor operated in photodiode mode with reverse bias of 72 V. The Passivated Implanted Planar Silicon (PIPS) detector from Mirion Technologies (PD 2000-40-300 AM) is housed in a tungsten shield to prevent hard X-ray contamination and has a graphite collimator in front, which restricts the acceptance angle to 28° full-angle, see Fig. 4. Similar detectors have been used in the past to detect fusion products on other magnetic fusion devices[23–25].

The signal pulse arises when a photon or charged particle enters the depletion region of the semiconductor and creates a population of electron-hole pairs. These charge carriers give rise to a current that is collected and amplified with a transimpedance amplifier with a 2 MHz corner frequency (Femto HCA-2M-1M-C) external to the vacuum vessel. We are interested here in registering MeV alpha particles, but

photons and lower energy particles can also induce signal, so a main driver in the design was to avoid this signal contamination. This was accomplished through two interventions. First, the detector is oriented such that there is no direct line-of-sight from the core plasma to the detector; we rely on the ~3 T magnetic field of the LHD to steer the alpha particles to the detector. Lorentz orbit tracing calculations[22] were carried out to determine the orientation of the detector that maximizes alpha particle collection without directly facing the plasma. Second, to block scattered photons and lower energy photons emitted from the divertor plasma, a 2 μm thick platinum foil was placed between the collimator and the detector, effectively shielding photons below ~4 keV while having a minimal effect on the alpha particles themselves, which have range of 5.9 μm in Pt at 4 MeV.

The detector was installed on a movable manipulator that can be inserted up into the LHD divertor plasma from the 10.5-L diagnostic port[28].

As the electrical signal must be transported over 9 m of single conductor cable before the first amplification stage, signal attenuation tests were conducted on a test stand with a [210]Po alpha particle source ($E = 5.41$ MeV). It was found that due to the capacitance of the biased detector ($C = 25$ nF), pulse attenuation only becomes significant when the parasitic capacitance exceeds 5 nF, much less than the capacitance of the 9 m cable run. To address the impact of the cable inductance, the first part of the experimental run was dedicated to minimizing sources of stray electric and magnetic fields during the measurement.

In order to protect the detector from the 1 MW/m² divertor heat flux during the 10 s long LHD pulses, a graphite heat shield was designed (see Fig. 2) using the ANSYS© finite element program to model thermal performance. DS-4 graphite with a heat conductivity $K = 105$ W/mK was chosen for the heat shield and collimator. It was determined that conductive cooling through the body to the water-cooled stage and radiative cooling would be sufficient to maintain the semiconductor at temperatures below its 100 °C operating limit. This was verified with thermocouple measurements during the experimental run.

### Signal processing

The small current signal from the detector is amplified with a transimpedance amplifier (Femto, model HCA-2M-1M-C) with corner

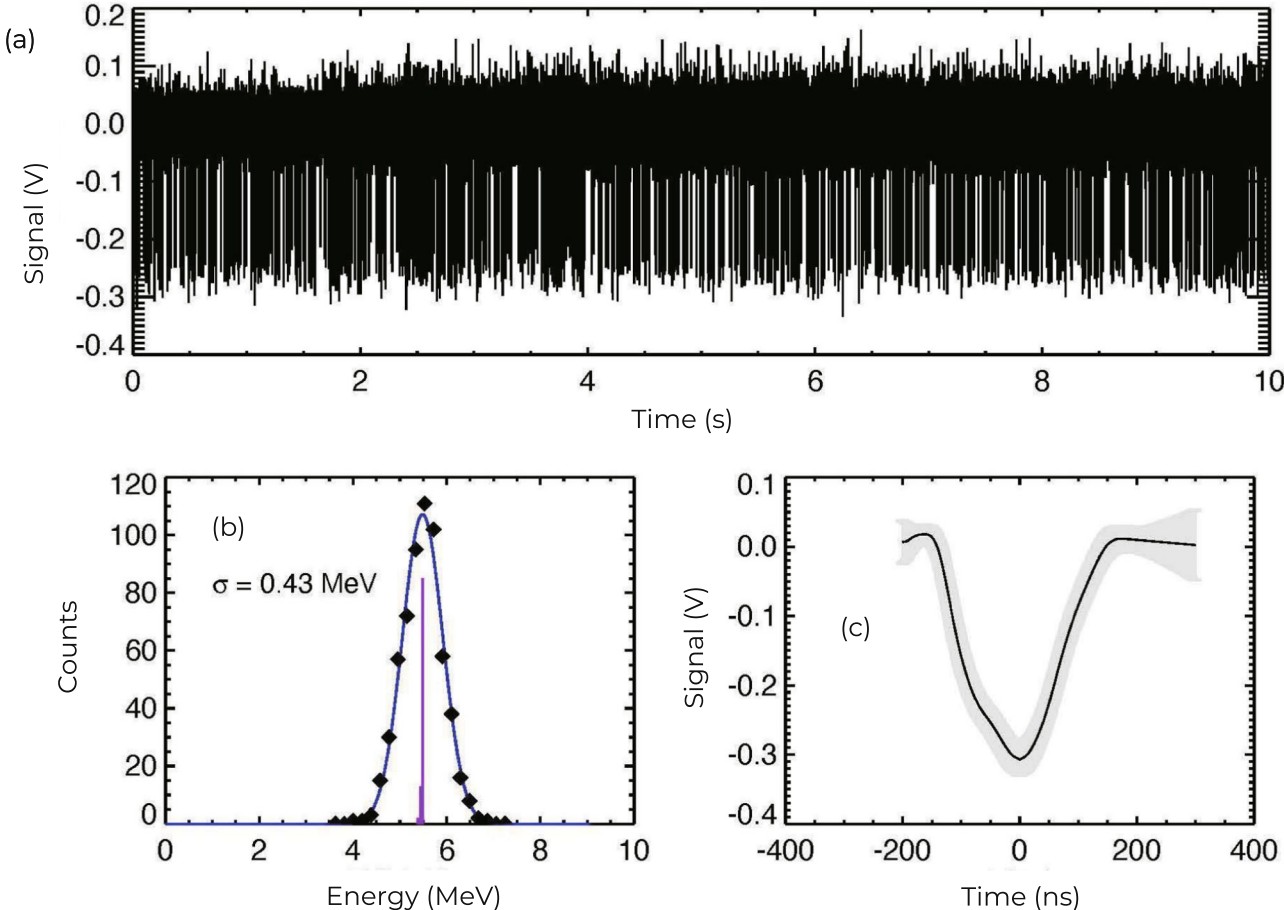

**Fig. 5 | In situ calibration of the alpha particle detector with a ²⁴¹Am alpha particle source. a** Time history of ²⁴¹Am calibration data taken in situ showing the fast, mono-energetic, negative pulses. **b** A histogram of pulse heights, converted to energy using the known energies of the ²⁴¹Am alpha particles. The energy resolution of the detector σ is 0.43 MeV at 5.486 MeV, better than 10%. The purple lines indicate the relative strengths of the energy components of the source. **c** An average of 609 pulses from the ²⁴¹Am alpha particle source digitized at 250 MHz (Techno AP, APV8508). The gray error bars represent the standard deviation of the mean.

frequency 2 MHz and gain $1 \times 10^6$ V/A. The analog data signal is then digitized at 10 MHz (National Instruments, model PXI-6115) and the pulses counted in post processing.

The pulse counting algorithm is a peak detector and pulse shape discriminator. First, any saturated segments of the data record are flagged. This can occur when the detector floats with the plasma potential relative to the ground of the digitizer outside of the input range. The signal is then passed through a high pass filter with cutoff frequency $f_c = 100$ Hz, and the local maxima are located. On the second pass, the points around those peaks are compared to the pulse shape obtained from a high-resolution calibration with a ²⁴¹Am source, described in the next section. If the residual error is smaller than a critical value (root mean square error < 0.2), the pulse is counted. Pulse shape discrimination results in a rejection rate of up to 50% in the noisy plasma environment.

## Calibration

In order to calibrate the detector, a 0.015 μCi (560 Bq) ²⁴¹Am source, which has a primary decay channel to neptunium by emitting a 5.486 MeV alpha particle, was mounted directly to the detector cap with the detector installed on the movable manipulator. The signal acquisition instrumentation and cabling were therefore the same in both the calibration and plasma runs, key to verifying pulse shape.

The results of the calibration are shown in Fig. 5. The pulses are collected at a rate of about 100 s⁻¹ and digitized at a rate of 250 MHz.

(In the experiment, where long data records are required, the data is digitized at a lower rate of 10 MHz, and we use the high time-resolution pulse for pulse shape discrimination.) As shown in Fig. 5(c), they have an average FWHM ~175 ns and average amplitude of 300 mV. The amplitude can be directly related to the energy of the particle. The primary energy (85% of decays) of the alpha is 5.486 MeV, giving a calibration factor of 54.7 mV/MeV. The histogram in Fig. 5 (b) shows the energy resolution to be 0.43 MeV, or about 8%. Note that the 2 μm foil was left in place for the calibration, so at least a portion of the energy spread is due to straggling in the foil.

## Data availability

The LHD data and processed data used in the Figures within can be accessed from the LHD data repository at https://www-lhd.nifs.ac.jp/pub/Repository_en.html. Email corresponding author for access instructions.

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

## Acknowledgements
S.O. would like to acknowledge critical early conversations with the late Prof. W.A. Cooper which served as the inspiration to pursue p$^{11}$B fusion experiments on LHD. The TAE team would like to thank their shareholders for their support and trust and the entire TAE and NIFS staffs for their dedication, excellent work, and extra efforts.

## Author contributions
R.M. conceived of the diagnostic and its conceptual design, conducted the analysis of the experimental data, and wrote the majority of the paper. K.O. installed and commissioned the diagnostic, collected data, and ran supporting simulations. T.T. and H.G. organized the collaboration and edited the manuscript. I.A. and V.K. designed and tested electronics and conducted preliminary experiments. P.M. designed the diagnostic support structure and shielding. S.O. led the LHD experiments and advised on experimental design. M.I. supported experiments on the LHD. S.K. assisted in experimental design and data analysis. H.N. and M.S. assisted in interfacing diagnostic with the LHD. S.Z. conducted FEM calculations of heat transport in diagnostic housing and advised on shielding. M.B. orchestrated lab to lab MOU. M.O. consulted on data analysis and manuscript.

## Competing interests
TAE Technologies is a private corporation owned and financially supported by its shareholders. The TAE authors of this manuscript (R.M., T.T., I.A., H.G., P.M., S.Z., M.B) may have a financial interest in the company. The remaining authors declare no competing interests.
