## [Peer Review File · Nature Communications]

First measurements of p11B fusion in a magnetically confined plasmaREVIEWER COMMENTS

Reviewer #1 (Remarks to the Author):

This is a potentially exciting result showing a measurement of p-B11 reaction products from a magnetically confined plasma. If confirmed, it is noteworthy and publication in this journal would be appropriate.

I am inclined to believe the measurements shown, but I do have concerns. With great claims comes great responsibility to defend those claims, and I feel the authors should do more to make their measurements definitive.

The key measurement rests on signals measured using a silicon diode. As the authors note, these detectors are not only sensitive to alpha particles but also photons and other particles that might create electron-hole pairs in the silicon. The authors, being aware of this, took some pains to remove a direct line of sight between the plasma and this detector as well as some shielding to remove the lowest-energy x-ray photons <4 keV. Nonetheless, I remain concerned that the authors have not ruled out a particle/x-ray background as a possible cause of the signal.

It is promising that, as shown in Figure 2, the signals disappear when there is no boron powder injection. However, the complete removal of the injection also removes all fast particles of any type that might produce a fast-particle or x-ray background. I feel that a third test is warranted, namely a test where the NB was used to inject something other than boron into the plasma, rather than simply turning on and off the NB injection altogether. Such a test would still have high energy particles entering the plasma that might be producing a particle/x-ray background, but if the injected beam does not contain boron, there should be no alpha particles and the signal will hopefully still look like Figure 2d.

It is likewise promising that the signal-to-simulation comparison of Figure 3c has a similar shape. Nonetheless, that shape is in proportion to the amount of beam particles being injected into the plasma (i.e., the measurement in Figure 3c is proportional to the shape in Figure 3b). That could mean that it is just a signature of a particle/x-ray background rather than a true alpha-particle signature. The predicted spike at 5.3 s is a feature unique to the presence of alpha particles, since it is expected from the p-B11 resonance. Had it been observed by the authors, that would have been definitive. Its absence is troubling to me, and in its absence and in the absence of the null test I described above, I feel that the measurements are promising but not definitive.

In summary, I feel an additional null test is warranted (data from a shot with NB injection but no boron) for this result to be compelling in the absence of a truly unique p-B11 alpha particle signature (e.g., the predicted spike at 5.3 s in Figure 3c). Without such additional data, I can't recommend this article be accepted in its present form. I believe this test is achievable and encourage the authors to complete such a test, update the manuscript, and resubmit.

Some additional minor remarks;

* The introduction might benefit from a reading by a non-magnetic-confinement fusion colleague of the authors. It is well referenced and mostly understandable, but there is some jargon that someone from, say, inertial confinement fusion, might not readily understand.

* On page 3, "The effects of fast ion velocity space diffusion..." is one such example of jargon. I'm assuming the authors are referring to changes in the tail of the ion particle velocity distribution that is responsible for the p-B11 reactions, but I don't understand the reference to "diffusion" in that context.

* Figure 1 caption: Please check but I think the Left/Right descriptions are flipped from what is in the graphic.

Reviewer #2 (Remarks to the Author):

In this manuscript, the authors report experiments done on the LHD in which proton beams interacting with a boron-containing plasma produced p11B fusion reactions. This is, as far as I am aware, the first report of p11B reactions in a magnetically confined plasma. The authors compare their experimental results with numerical calculations of expected p11B yields.

The manuscript is well-written, it is interesting, and the results being reported are novel. Moreover, I think this will be seen in the plasma physics community as an important first; there has been substantial interest (especially recently) in the p11B reaction, and demonstrating it in this kind of device is significant (more so if the authors are successful in pursuing some of the follow-up research they suggest in the manuscript). However, I do have a few questions and concerns.

1. In the comparison of the experimental results with the numerical calculations, the validation consists of observing that both have ramp-ups in the number of fusion events that track with the initial increase in boron density and that both have sharp drops in the number of fusion events when the neutral beams shut off. These features provide nice evidence that the signal is from p11B reactions. But could these features of the data not have been predicted equally well without any need for numerical calculations? Actually, I would have said that the really interesting physics content of the numerical curve in Figure 3

is the discrepancy between the numerical calculation and the experimental results. As the authors note in Section V, some parts of this discrepancy could have significant implications.

2. I am not a diagnostician, but as far as I can tell from the manuscript, the authors do not have an expected mapping between the magnitude of the experimental alpha particle signal and the total number of fusion events in the system (that is, without assuming that the numerics are correct). Their numerical results suggest a number of events, but if, for example, the numerics were all too high or too low by some constant factor, this would not necessarily be clear from the experimental data. This makes the statement in the abstract that the measurements have "confirmed yields of $\sim 10^{13}$ " seem too strong. Am I missing something there? One way or another, this is a point that I would appreciate seeing clarified in the manuscript.

I also have four somewhat more minor questions and comments.

3. The authors provide a physical explanation for the spike in the numerically predicted fusion yield at about 5 s, but the numerical curve has some other structures as well. It would be interesting to know if the rest (for example, the spike shortly after 7 s) can be explained.

4. The first sentence of the second paragraph of Section VI states that: "For the last several decades, the study of the p11B fusion reaction has been confined to theoretical reactor studies and nuclear physics experiments at particle accelerators." Elsewhere in the paper the authors state that it has also been studied in laser experiments. Why the discrepancy?

5. There is a minor typo in the caption of Figure 2: "boron powder injection.."

6. The DOI link in Ref. [22] is broken. (The printed URL is fine, but one of the underlying hyperlinks has a typo).

Reviewer #1:

This is a potentially exciting result showing a measurement of p-B11 reaction products from a magnetically confined plasma. If confirmed, it is noteworthy and publication in this journal would be appropriate.

I am inclined to believe the measurements shown, but I do have concerns. With great claims comes great responsibility to defend those claims, and I feel the authors should do more to make their measurements definitive.

Thank you. We agree that this is a noteworthy result and do not take the associated responsibility lightly. We hope the discussion below will convince you, as we've convinced ourselves, that the measurement is sound.

The key measurement rests on signals measured using a silicon diode. As the authors note, these detectors are not only sensitive to alpha particles but also photons and other particles that might create electron-hole pairs in the silicon. The authors, being aware of this, took some pains to remove a direct line of sight between the plasma and this detector as well as some shielding to remove the lowest-energy x-ray photons <4 keV. Nonetheless, I remain concerned that the authors have not ruled out a particle/x-ray background as a possible cause of the signal.

It is promising that, as shown in Figure 2, the signals disappear when there is no boron powder injection. However, the complete removal of the injection also removes all fast particles of any type that might produce a fast-particle or x-ray background. I feel that a third test is warranted, namely a test where the NB was used to inject something other than boron into the plasma, rather than simply turning on and off the NB injection altogether. Such a test would still have high energy particles entering the plasma that might be producing a particle/x-ray background, but if the injected beam does not contain boron, there should be no alpha particles and the signal will hopefully still look like Figure 2d.

It is indeed true that the detector is sensitive to x-rays and lower energy particles, as well as MeV alphas, so we shielded the detector geometrically and with a 2 um Pt foil and took great care to confirm the origin of the signal. In fact, we have already conducted the test suggested.

In the experiment, the boron is dropped into the plasma as a sub-millimeter powder under the force of gravity alone, not as a high energy beam. The boron is then ionized by the plasma and subsumed to become another plasma ion species. The neutral beams inject protons only. Figure 2d therefore shows the case the Reviewer wishes to see. In this shot, there was no powder drop, so there is relatively little boron (only the residual amount from previous shots), but there is still neutral beam injection. Therefore, there is a population of high energy

protons in the plasma, but no boron for them to fuse with. The absence of the pulses in Figure 2d shows that high energy protons alone do not induce the signal

We have added the sentence, “In both discharges, NB3 was fired from 3.3 s to 5.3 s and NB1 was fired from 5.3 to 7.3 s (as in Figure 3a), so there is steady neutral beam injection during the window shown.” to make this clear.

It is likewise promising that the signal-to-simulation comparison of Figure 3c has a similar shape. Nonetheless, that shape is in proportion to the amount of beam particles being injected into the plasma (i.e., the measurement in Figure 3c is proportional to the shape in Figure 3b). That could mean that it is just a signature of a particle/x-ray background rather than a true alpha-particle signature. The predicted spike at 5.3 s is a feature unique to the presence of alpha particles, since it is expected from the p-B11 resonance. Had it been observed by the authors, that would have been definitive. Its absence is troubling to me, and in its absence and in the absence of the null test I described above, I feel that the measurements are promising but not definitive.

Similar to the above, since the boron is not injected as a high energy beam, the similar shape of the alpha detector signal to the boron density could not be due to a fast particle or x-ray background.

The fusion rate is given by,

$$R dV = n_p n_B \sigma(v_p) v_p dV$$

where n_p is the proton density, n_B is the boron density, σ is the fusion cross section and v_p is the velocity of the protons (we assume $v_p \gg v_b$ so the relative velocity, $v_{rel} = v_p - v_b \sim v_p$). The neutral beam injected fast ions reach equilibrium in about 100 ms, so we should actually expect the leading edge of the detector signal to mirror the dynamics of the boron density, as it does.

At the trailing edge we see the case complementary to Figure 2d: boron present, high energy beam absent. It can be seen that when NB1 turns off at $t=7.3$ s, there is a fast drop in the detector signal even though the boron population persists. Affirming that both high energy protons and boron are necessary for the signal, and the either by itself produces no signal.

Regarding the burst in the calculated rate at $t=5.3$ s: we have been investigating this since our original submission and have discovered an issue with an interpolation subroutine that arises when there are fast changes to NB parameters. This subroutine was used in the calculation and was exaggerating the size of the peak at $t=5.3$ s.

The measurement of the NB energy was being interpolated inaccurately which led to an inflation of 2-3 keV around discrete bursts. Because the burst at $t=5.3$ s occurs when the NB energy is near the pB11 resonance, this small error in beam energy had a large effect on the calculated fusion rate.

The corrected plot has been inserted in place of the previous. Now we see that both the measured and calculated rates register a similar jump in signal when NB1 turns on at $t=5.3$ s. Although it looks like the calculated rate might decay more quickly than the measured rate, the two signals agree within their error bars.

In summary, I feel an additional null test is warranted (data from a shot with NB injection but no boron) for this result to be compelling in the absence of a truly unique p-B11 alpha particle signature (e.g., the predicted spike at 5.3 s in Figure 3c). Without such additional data, I can't recommend this article be accepted in its present form. I believe this test is achievable and encourage the authors to complete such a test, update the manuscript, and resubmit.

Since we have in fact conducted the null test requested and the agreement is now much improved in Figure 3c, we hope that your concerns have been assuaged.

Thank you very much for the time and effort spent on this careful review.

Some additional minor remarks;

* The introduction might benefit from a reading by a non-magnetic-confinement fusion colleague of the authors. It is well referenced and mostly understandable, but there is some jargon that someone from, say, inertial confinement fusion, might not readily understand.

We have had non-experts read the manuscript and haven't uncovered any non-referenced jargon, other than the instance below. If the Reviewer has other specific instances, we can re-work.

* On page 3, "The effects of fast ion velocity space diffusion..." is one such example of jargon. I'm assuming the authors are referring to changes in the tail of the ion particle velocity distribution that is responsible for the p-B11 reactions, but I don't understand the reference to "diffusion" in that context.

We have modified the sentence to remove the term. It now reads, "The effects of the resonances in the $p^{11}\text{B}$ cross section in the presence of high energy tails and non-thermal distributions on the fusion rate can also be explored."

* Figure 1 caption: Please check but I think the Left/Right descriptions are flipped from what is in the graphic.

Yes, they were, thank you. Caption has been corrected.

Reviewer #2:

In this manuscript, the authors report experiments done on the LHD in which proton beams interacting with a boron-containing plasma produced p11B fusion reactions. This is, as far as I am aware, the first report of p11B reactions in a magnetically confined plasma. The authors compare their experimental results with numerical calculations of expected p11B yields.

The manuscript is well-written, it is interesting, and the results being reported are novel. Moreover, I think this will be seen in the plasma physics community as an important first; there has been substantial interest (especially recently) in the p11B reaction, and demonstrating it in this kind of device is significant (more so if the authors are successful in pursuing some of the follow-up research they suggest in the manuscript). However, I do have a few questions and concerns.

Thank you for the thoughtful reading and for the positive endorsement. We agree with the expectation that this will be viewed as an important first. We hope to answer your questions and alleviate your concerns below.

1. In the comparison of the experimental results with the numerical calculations, the validation consists of observing that both have ramp-ups in the number of fusion events that track with the initial increase in boron density and that both have sharp drops in the number of fusion events when the neutral beams shut off. These features provide nice evidence that the signal is from p11B reactions. But could these features of the data not have been predicted equally well without any need for numerical calculations? Actually, I would have said that the really interesting physics content of the numerical curve in Figure 3 is the discrepancy between the numerical calculation and the experimental results. As the authors note in Section V, some parts of this discrepancy could have significant implications.

Since our original submission we have discovered an issue in an interpolation subroutine used in the calculation that was exaggerating the size of the peak at $t=5.3$ s.

The measurement of the NB energy was being interpolated inaccurately when first derivatives were large which led to an inflation of 2-3 keV around discrete bursts. Because the burst at $t=5.3$ s occurs when the NB energy is near the pB11 resonance, this small error in beam energy had a large effect on the calculated fusion rate.

The corrected plot has been inserted in place of the previous and we have edited the accompanying text. Now we see that both the measured and calculated rates register a similar jump in signal when NB1 turns on at $t=5.3$ s. Although it looks like the calculated rate might decay more quickly than the measured rate, the two signals agree within their error bars.

To the referee's larger point that the common features of the curves called out in the text could have been predicted without simulation, this is probably true, at least in a qualitative sense. But the leading edge of the rate curves is actually slightly different from the leading edge of the boron emission curve due to profile dynamics, and the decay rate involves several fast ion

processes, including slowing down, charge exchange, and scattering, which would be difficult to capture accurately without the simulation.

2. I am not a diagnostician, but as far as I can tell from the manuscript, the authors do not have an expected mapping between the magnitude of the experimental alpha particle signal and the total number of fusion events in the system (that is, without assuming that the numerics are correct). Their numerical results suggest a number of events, but if, for example, the numerics were all too high or too low by some constant factor, this would not necessarily be clear from the experimental data. This makes the statement in the abstract that the measurements have "confirmed yields of $\sim 10^{13}$ " seem too strong. Am I missing something there? One way or another, this is a point that I would appreciate seeing clarified in the manuscript.

This is a good point. We have attempted an absolute comparison between the experimentally measured local rate and the calculated global rate, but they currently differ by a factor of several, such that the calculated rate $>$ measured rate. There are many known uncertainties, most notably the fast ion confinement time of the plasma, the boron density, and the fraction of emitted alphas that reach the detector. While the first two of these might be constrained with additional measurements, the last is particularly problematic.

The fraction of alphas which reach the detector is a function of the energy spectrum of the emitted particles. Since we can only measure the energy spectrum of the particles which arrive to the detector, we can't measure the source spectrum, leading to severe survivorship bias. Future work will attempt to invert the problem. For now, we have removed the statements about confirmed yield from the abstract.

I also have four somewhat more minor questions and comments.

3. The authors provide a physical explanation for the spike in the numerically predicted fusion yield at about 5 s, but the numerical curve has some other structures as well. It would be interesting to know if the rest (for example, the spike shortly after 7 s) can be explained.

The spike at $t = 7$ s was also an artifact of inaccurate interpolation. We now see spikes of comparable size in measurement and simulation near $t = 7$ s.

4. The first sentence of the second paragraph of Section VI states that: "For the last several decades, the study of the p11B fusion reaction has been confined to theoretical reactor studies and nuclear physics experiments at particle accelerators." Elsewhere in the paper the authors state that it has also been studied in laser experiments. Why the discrepancy?

An oversight. Laser experiments (aka IFE) has been added to Section IV. Thank you.

5. There is a minor typo in the caption of Figure 2: "boron powder injection.."

Double period removed from Figure 2 caption, thank you

6. The DOI link in Ref. [22] is broken. (The printed URL is fine, but one of the underlying hyperlinks has a typo).

Thank you for notification of the broken DOI link. It appears when the file is compiled the address is being linked to an arXiv address instead of the URL as written. I will have to work with the editors on this one.

REVIEWER COMMENTS

Reviewer #1 (Remarks to the Author):

The authors did a good job of carefully addressing both my remarks and those of the other reviewer. To my embarrassment, as I was composing my review, I clearly lost track of the geometry (i.e., proton injection and boron powder drop), and agree with the authors that the data is definitive for the reasons they outline.

While I know that the references include nice pictures of the overall experiment, one suggestion might be to include a small diagram of the overall experimental configuration. I'm a visual person by nature and greatly appreciate such diagrams to quickly orient me toward what was done. This could perhaps be done as part of Figure 1 so it was clearer where the detector was relative to the full experimental geometry? I make this as a suggestion, not a requirement.

I recommend this manuscript be accepted for publication in this journal.

Reviewer #2 (Remarks to the Author):

The authors' response does address my main questions about the manuscript. However, some of the explanations they gave in the response to my comments really ought to be clarified in the text itself. In particular, the point about the numerical calculation and experimental data not being comparable in absolute terms (only in relative shape) does not, as far as I can see, appear anywhere in the manuscript apart from the caption of Figure 3. It seems like an important caveat. Without more direct discussion of the issue in the text, some of the descriptions of these results (for example, the remaining language in the abstract about "very good agreement with calculations of the global rate") may be misleading. This is not to say that I have any objection to the numerical validation they are doing, just that I think it needs to be clear exactly what comparison is being made.

If the authors can clarify the above point, I would recommend publication.

I would also suggest that the manuscript would be clearer if the authors included some of the explanation they sent me as to why the numerical results give information that could not have easily been found analytically (i.e., something along the lines of the last sentence from their response to point

1). However, this may be a matter of personal preference, and I would not object if the authors would prefer to leave it out. I do find the numerical comparison substantially more compelling in its new, corrected form.

Reviewer #1:

The authors did a good job of carefully addressing both my remarks and those of the other reviewer. To my embarrassment, as I was composing my review, I clearly lost track of the geometry (i.e., proton injection and boron powder drop), and agree with the authors that the data is definitive for the reasons they outline.

Not at all, if you lost track it would happen to many other readers as well. We hope the additional text will help prevent this.

While I know that the references include nice pictures of the overall experiment, one suggestion might be to include a small diagram of the overall experimental configuration. I'm a visual person by nature and greatly appreciate such diagrams to quickly orient me toward what was done. This could perhaps be done as part of Figure 1 so it was clearer where the detector was relative to the full experimental geometry? I make this as a suggestion, not a requirement.

It's a great suggestion. We've added another image to Figure 1 showing the LHD vacuum vessel with the approximate location of the measurement and updated the caption appropriately.

I recommend this manuscript be accepted for publication in this journal.

Thank you for the recommendation.

Reviewer #2:

The authors' response does address my main questions about the manuscript. However, some of the explanations they gave in the response to my comments really ought to be clarified in the text itself. In particular, the point about the numerical calculation and experimental data not being comparable in absolute terms (only in relative shape) does not, as far as I can see, appear anywhere in the manuscript apart from the caption of Figure 3. It seems like an important caveat. Without more direct discussion of the issue in the text, some of the descriptions of these results (for example, the remaining language in the abstract about "very good agreement with calculations of the global rate") may be misleading. This is not to say that I have any objection to the numerical validation they are doing, just that I think it needs to be clear exactly what comparison is being made.

We have added some additional language to make more clear the point that the comparison between the simulation and measurement is relative in this work.

- In the abstract, "the measured rate shows very good *relative* agreement with calculations of the global rate"

- We've modified the end of Section V to read: "An absolute comparison between the measurement and calculation requires knowledge of the alpha energy spectrum at the source (i.e., in the plasma) for accurate orbit tracing, while we can only measure the energy spectrum of those alphas that arrive at the detector, creating severe survivorship bias. We therefore rely on the dynamical, relative agreement as further confirmation that the measured signal is indeed $p^{11}B$ fusion born alpha particles."

If the authors can clarify the above point, I would recommend publication.

Thank you for the recommendation

I would also suggest that the manuscript would be clearer if the authors included some of the explanation they sent me as to why the numerical results give information that could not have easily been found analytically (i.e., something along the lines of the last sentence from their response to point 1). However, this may be a matter of personal preference, and I would not object if the authors would prefer to leave it out. I do find the numerical comparison substantially more compelling in its new, corrected form.

We have added the sentence, "While the calculation of the fusion rate is straightforward, capturing the fast ion dynamics and the resulting fast ion profile and energy spectrum requires modeling" to the end of the first paragraph of Section V.

REVIEWERS' COMMENTS

Reviewer #2 (Remarks to the Author):

The authors have fully addressed my concerns. I recommend this manuscript for publication.